# The Associations between Physical Activity, Functional Fitness, and Life Satisfaction among Community-Dwelling Older Adults

**DOI:** 10.3390/ijerph19138043

**Published:** 2022-06-30

**Authors:** Shih-Huei Syue, Hui-Fei Yang, Cheng-Wei Wang, Shih-Yu Hung, Pei-Hsuan Lee, Sheng-Yu Fan

**Affiliations:** 1Department of Family Medicine, Ditmanson Medical Foundation Chia-Yi Christian Hospital, Chia-Yi 600, Taiwan; 07492@cych.org.tw; 2Department of Community Health, Ditmanson Medical Foundation Chia-Yi Christian Hospital, Chia-Yi 600, Taiwan; 05791@cych.org.tw (H.-F.Y.); 13649@cych.org.tw (C.-W.W.); 10664@cych.org.tw (S.-Y.H.); 13199@cych.org.tw (P.-H.L.); 3Institute of Gerontology, College of Medicine, National Cheng Kung University, No. 1, University Road, Tainan City 701, Taiwan

**Keywords:** mediator, physical exercise, physical fitness, subjective well-being

## Abstract

Previous studies showed physical activity had benefits for older adults’ life satisfaction, but the mechanism was unclear. This study aimed to investigate whether older adults with more physical activity had better functional fitness and life satisfaction, and whether functional fitness mediated the relationship between physical activity and life satisfaction. A cross-sectional study design was employed, and 623 older adults (73.71 ± 5.91 years) were recruited. Physical activity, functional fitness, life satisfaction, and demographic characteristics were collected. Compared with older adults with low physical activity, those with high (*B* = 0.41, *p* = 0.025) and moderate (*B* = 0.40, *p* = 0.041) physical activity had better life satisfaction; those with high physical activity had better lower limb muscle strength (*B* = 1.71, *p* = 0.001), upper (*B* = 2.91, *p* = 0.032) and lower (*B* = 3.12, *p* = 0.006) limb flexibility, cardiorespiratory endurance (*B* = 6.65, *p* = 0.008), and dynamic balance ability (*B* = −1.12, *p* < 0.001). Functional fitness did not mediate the relationship between physical activity and life satisfaction. Promoting physical activity may be useful for increasing older adults’ functional fitness and life satisfaction, but the effects on functional fitness only occurred at a high level of physical activity, and the effect of physical activity on life satisfaction was not mediated by functional fitness.

## 1. Introduction

In Taiwan, older adults will comprise 20% of the whole population by 2025, resulting in a super-aged society [1]. With the rise in this population, health promotion is an important issue [2]. Physical activity is one of the many strategies with which to improve older adults’ health, and it has physical and psychological benefits for older adults [3].

Life satisfaction is a subjective evaluation of a person’s quality of life [4]. Older adults with more physical activity have better life satisfaction [5,6]. However, the mechanism behind the link between physical activity and life satisfaction was unknown. Blair, Cheng, and Holder [7] presented a model in which there are pathways from physical activity to health outcomes through functional fitness. Physical activity can improve functional fitness and then improve health outcomes, such as health or quality of life [7] (see Figure 1).

Functional fitness means the physiological capacity to perform daily activities safely and independently, including muscle strength, cardiorespiratory endurance, flexibility, and balance ability [8]. Previous studies showed positive relationships between physical activity and functional fitness. Older adults with higher levels of physical activity tended to have better upper [9,10,11,12,13,14] and lower [9,10,11,12,14,15] limb muscle strength, upper [10,11,12,14,16] and lower [9,10,11,14] limb flexibility, cardiorespiratory endurance [9,10,11,12,13,14,15,17], static balance ability [17], and dynamic balance ability [9,10,11,12,13,14,17,18]. However, some studies state the opposite, claiming that there are no relationships between physical activity and upper limb [16,17,19] and lower [13,16,17,19] limb muscle strength, upper [9,13,15,19] and lower [13,15,16,19] limb flexibility, cardiorespiratory endurance [13,18,19], and dynamic balance ability [16,19].

Furthermore, some studies have shown dose–response relationships between physical activity and functional fitness [10,20]. However, de Melo et al. [9] found that only older adults with high physical activity had better functional fitness than those with low physical activity, rather than those with moderate physical activity.

Regarding the relationships between physical activity, life satisfaction, and functional fitness, the model assumed a mediating role for functional fitness. While considering physical activity and functional ability at the same time, these two variables were significantly related to life satisfaction [21]. Older adults with high physical activity had not only better functional ability and activities of daily living, but also higher life satisfaction and functional autonomy [22]. However, few studies have investigated the relationships empirically. A quantitative study was conducted by using standardized measurements. The international physical activity questionnaire (IPAQ) short form covers mild, moderate, and vigorous levels of physical activity and is suitable for assessing the physical activity levels of older adults [23,24]. The senior fitness test battery [8] was used to collect objective data on functional fitness.

Understanding the role of functional fitness in the link between physical activity and life satisfaction is useful for establishing a theoretical model, as well as supporting functional fitness training for older adults in clinical care. Therefore, this study aimed to explore the relationships between physical activity and functional fitness and life satisfaction, and whether there was a dose–response relationship, as well as the mediating role of functional fitness therein. The hypotheses were that older adults with higher physical activity had better functional fitness and life satisfaction, and the effect of physical activity on life satisfaction was mediated through functional fitness.

## 2. Methods

### 2.1. Participants and Study Design

A cross-sectional study design was used. The inclusion criteria for the participants were older adults, aged 65 years or more, and able to take functional fitness assessments and complete questionnaires.

The participants were recruited from 12 functional fitness and health screening stations held by the Sports Administration of the Ministry of Education in Chia-Yi, a city in Southern Taiwan. The period of data collection was from January 2016 to November 2017. Adults who were not suitable in terms of functional fitness as assessed by the Physical Activity Readiness Questionnaire were excluded [25]. Ethical approval was obtained from the institutional review board of the Ditmanson Medical Foundation Chia-Yi Christian Hospital (IRB number: 2020024).

### 2.2. Measurements

The demographic characteristics included age, sex, educational levels, marital status, and living accommodation status (alone/with family).

Physical activity: The IPAQ short form was used to assess the physical activity levels of the older adults [26]. Three intensities of physical activities during the last seven days were collected, including (1) vigorous-intensity activities such as speed swimming or playing basketball, (2) moderate-intensity activities such as slow dancing or strenuous household chores, and (3) low-intensity activities such as walking. Each activity had its own metabolic equivalent (MET) energy expenditure, and the vigorous-, moderate-, and low-intensity activities equaled 8.0, 4.0, and 3.3 METs. The amount of physical activity was estimated by weighting the intensity and time spent performing each activity with its MET. Older adults were categorized into groups as having low physical activity (LPA, 0 ≦ METs < 600), moderate physical activity (MPA, 600 ≦ METs < 3000), and high physical activity (HPA, METs ≦ 3000) [26]. The IPAQ has been translated into a Taiwanese version with good reliability and validity [27].

Functional fitness: A test battery was conducted to assess older adults’ functional fitness [8]. (1) The arm curl test was used to measure upper limb muscle strength: the number of times (frequency) arm curls with the weight (male, 8 pounds, and female, 5 pounds) could be completed in 30 s. (2) The chair stand test was used to measure lower limb muscle strength: the number of times (frequency) older adults went from the sitting position to standing up and then back down in 30 s. (3) The back scratch test was used to measure upper limb flexibility: the distance (cm) between two fingers when one arm was behind the head and back over the shoulder and the other arm was behind the back. (4) The chair sit and reach test was used to measure lower limb flexibility: the distance (cm) between the tip of the fingertips and the toes when one foot remained flat and the other leg’s knee was straight. (5) The 2-minute step test was used to measure cardiorespiratory endurance: the number of full steps in 2 min (frequency), raising each knee to a point midway between the patella and iliac crest (top hip bone). The score was the number of times the right knee reached the required height. (6) The single leg stand test was used to measure static balance ability: the amount of time (in seconds) for which one could stand on one leg. (7) The seated up-and-go test was used to measure dynamic balance ability: the amount of time (in seconds) required to get up from a seat, walk 8 feet, and return to the seat.

Life satisfaction: A single item with an 11-point scale from 0 to 10 was used to evaluate life satisfaction: “In general, how satisfied are you with your life?”. A higher score indicated better life satisfaction [28]. The item had good convergent and divergent validity [29].

### 2.3. Statistical Analysis

Descriptive statistics are used to present the demographic characteristics of the participants. Spearman’s rho was used to test the correlations between physical activity and functional fitness and life satisfaction. Pearson correlations were used to test for correlations between functional fitness and life satisfaction.

PROCESS analysis [30] was used to test the pathways between physical activity, functional fitness, and life satisfaction. Physical activity was a categorical variable, including HPA, MPA, and LPA. Functional fitness, as a mediator, and life satisfaction, as a dependent variable, were the continuous variables. The covariate variables included age, sex, educational levels, married or not, and living accommodation status. PROCESS analysis could test three pathways at the same time, including physical activity to functional fitness, physical activity to life satisfaction, and physical activity to life satisfaction through functional fitness. It can reduce type I errors. SPSS 21.0 was used for statistical analysis, and *p* values less than 0.05 were taken as significant levels. The report of this study was based on the STROBE guidelines (the “strengthening the reporting of observational studies in epidemiology” statement) [31].

## 3. Results

A total of 623 older adults were recruited. The demographic characteristics of the participants are shown in Table 1. Physical activity was significantly related to life satisfaction, the arm curl test, the chair stand test, the chair sit and reach test, the 2 min step test, the single leg stand test, and the seated up-and-go test. Life satisfaction was significantly related to the chair stand test, 2 min step test, and seated up-and-go test (see Table 2).

The direct pathway from physical activity to life satisfaction when controlling for demographic characteristics and functional fitness was significant. Compared with older adults with LPA, and older adults with MPA, older adults with HPA had higher life satisfaction. Regarding the pathways from physical activity to functional fitness, compared with older adults with LPA, older adults with HPA had better performance in the chair stand test, back scratch test, chair sit and reach test, 2 min step test, and seated up-and-go test. Regarding the pathways from physical activity to life satisfaction through functional fitness, all the pathways were not significant (see Table 3). Figure 2 presents the pathways between physical activity, functional fitness, and life satisfaction.

## 4. Discussion

This study used data from functional fitness and health screening exams to explore the relationships between physical activity, functional fitness, and life satisfaction in older adults. Compared with older adults with LPA, those with HPA had better lower limb muscle strength, upper and lower limb flexibility, cardiorespiratory endurance, and dynamic balance ability. Older adults with HPA and MPA had higher life satisfaction than those with LPA. However, functional fitness did not play a mediating role between physical activity and life satisfaction.

The findings of this study support the idea that physical activity is positively correlated with most functional fitness categories, which is consistent with previous studies [9,10,11,12,14,32]. In addition, the findings revealed that the effects of physical activity on functional fitness only occurred with high physical activity, and there were no significant differences in functional fitness between MPA and LPA.

However, some studies showed dose–response effects of physical activity on activities of daily living and disability [33], as well as showing that older adults with high physical activity had the highest functional fitness, followed by those with moderate physical activity, and those with low physical activity had the lowest functional fitness [10,20]. Dondzila et al. [20] and Duncan et al. [10] used the steps a day as an index of physical activity. This study used the IPAQ to categorize the physical activity groups, and the standardized questionnaire includes different intensities of physical activities and can assess physical activity comprehensively. Furthermore, the finding was consistent with a previous meta-analysis showing that there was no dose–response relationship between physical activity and functional fitness [34]. Older adults need high-intensity physical activity to enhance functional fitness.

Similar to those of previous studies [5,6,32], the findings supported the idea that there was a potential benefit of physical activity for life satisfaction. In addition, both moderate and high levels of physical activity had potential benefits for life satisfaction. However, the findings did not support the mediating hypothesis about functional fitness. According to the correlation analysis, the lower limb muscle strength, cardiorespiratory endurance, and dynamic balance ability were significantly related to life satisfaction. Furthermore, after entering physical activity and functional fitness scores at the same time, the results only showed the physical activity as remaining significant. The influence of physical activity was larger than that of functional fitness on life satisfaction.

The findings refuted the mediating hypothesis for functional fitness. The potential explanations include the following: First, life satisfaction involves subjective evaluation about an individual’s quality of life [4], and older adults consider their whole lives rather than only physical health. Second, physical activity has psychological benefits, such as decreasing anxiety and depression and negative emotions [35,36], or increasing self-efficacy [35], autonomy [37], and mental health [38,39], and thus increased older adults’ life satisfaction. The pathway from physical activity to life satisfaction may pass through other psychological variables.

Regarding the clinical implications, older adults in community-dwelling settings can improve their functional fitness and life satisfaction by increasing physical activity. Even moderate physical activity can significantly increase life satisfaction. On the other hand, only high physical activity had positive effects on most functional abilities rather than only upper limb muscle strength and static balance ability. Clinical staff or caregivers can help older adults to achieve high levels of physical activity. Specific exercise training that focuses on the two abilities can be applied.

Some limitations of this study should be acknowledged. First, this study used a cross-sectional design, which cannot demonstrate cause and effect for the link between physical activity and functional fitness and life satisfaction. Second, some variables related to life satisfaction were not included, such as income, chronic diseases, exercise habits, and mental health. In addition, life satisfaction was measured with a single item, and a questionnaire with multiple items could be used for comprehensive measurement. Third, the data collection was conducted in functional fitness exam stations, and the participants all had mobility and physical conditions good enough for them to complete the functional fitness exam. The results cannot be generalized to older adults with disabilities or severe diseases. A future study could focus on older adults with sub-health statuses and collect other psychological variables to establish a model linking physical activity and life satisfaction.

## 5. Conclusions

In conclusion, physical activity benefits functional fitness and life satisfaction. The functional benefits were only observed among older adults with HPA, including improvements in lower limb muscle strength, upper and lower limb flexibility, cardiorespiratory endurance, and dynamic balance ability. Furthermore, older adults with HPA and MPA had better life satisfaction then those with LPA. The effects of physical activity on life satisfaction were not mediated by functional fitness.

## Figures and Tables

**Figure 1 ijerph-19-08043-f001:**
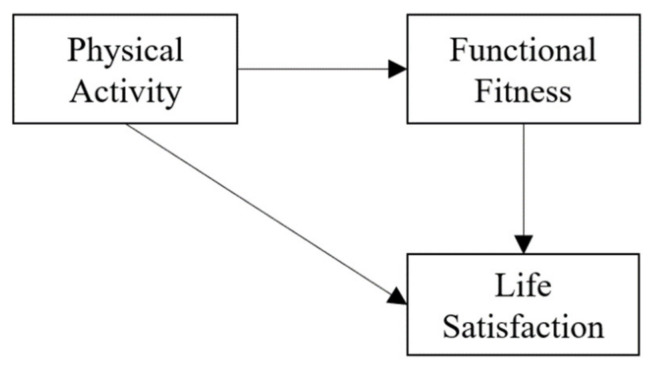
The conceptual model of this study.

**Figure 2 ijerph-19-08043-f002:**
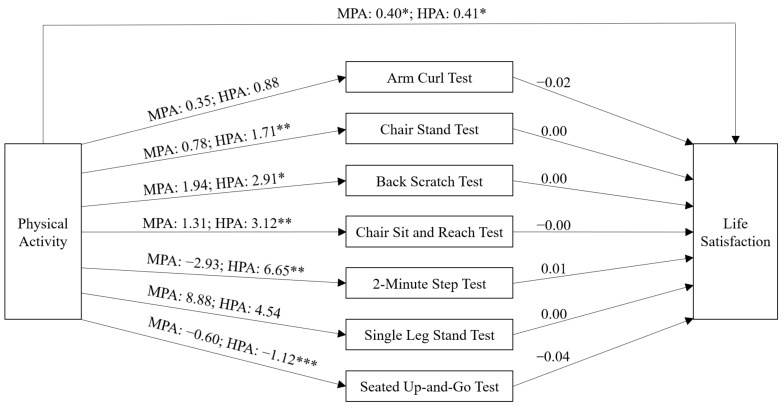
The pathways between physical activity, functional fitness, and life satisfaction. HPA: High Physical Activity; MPA: Moderate Physical Activity. * *p* < 0.05, ** *p* < 0.01, *** *p* < 0.001.

**Table 1 ijerph-19-08043-t001:** Demographic characteristics of the participants (N = 623).

Variables	N (%)
Age (years, M ± SD)	73.71 ± 5.91
Sex	
Male	248 (39.81%)
Female	375 (60.19%)
Educational levels	
Illiteracy and elementary school	274 (43.98%)
Junior high school	74 (11.88%)
Senior high school	127 (20.39%)
Undergraduate and postgraduate	148 (23.76%)
Marital status	
Married	479 (76.89%)
Others	144 (23.11%)
Living alone (Yes)	
Yes	130 (20.87%)
No	493 (79.13%)
Physical activity levels	
Low	168 (26.97%)
Moderate	179 (28.73%)
High	276 (44.30%)

**Table 2 ijerph-19-08043-t002:** The correlations between physical activity and life satisfaction and functional fitness.

	Physical Activity ^a^	Life Satisfaction ^b^
Life Satisfaction	0.14 ***	--
Arm Curl Test	0.10 *	0.03
Chair Stand Test	0.18 ***	0.13 **
Back Scratch Test	0.06	0.06
Chair Sit and Reach Test	0.10 *	0.07
2-Minute Step Test	0.17 ***	0.13 **
Single Leg Stand Test	0.17 ***	0.03
Seated Up-and-Go Test	−0.19 ***	−0.21 ***

^a^ Spearman’s rho, ^b^ Pearson correlation. * *p* < 0.05, ** *p* < 0.01, *** *p* < 0.001.

**Table 3 ijerph-19-08043-t003:** The pathways between physical activity, functional fitness, and life satisfaction.

Pathways	B (95% Confidence Interval)
Physical activity → Life Satisfaction (directly)	
MPA → Life Satisfaction	0.40 (0.02, 0.79) *
HPA → Life Satisfaction	0.41 (0.05, 0.77) *
Physical Activity → Functional Fitness	
MPA → Arm Curl Test	0.35 (−0.92, 1.62)
HPA → Arm Curl Test	0.88 (−0.29, 2.05)
MPA → Chair Stand Test	0.78 (−0.33, 1.90)
HPA → Chair Stand Test	1.71 (0.67, 2.75) **
MPA → Back Scratch Test	1.94 (−0.94, 4.82)
HPA → Back Scratch Test	2.91 (0.25, 5.57) *
MPA → Chair Sit and Reach Test	1.31 (−1.09, 3.71)
HPA → Chair Sit and Reach Test	3.12 (0.90, 5.33) **
MPA → 2-Minute Step Test	−2.93 (−8.27, 2.41)
HPA → 2-Minute Step Test	6.65 (1.72, 11.59) **
MPA → Single Leg Stand Test	8.88 (−0.71, 18.41)
HPA → Single Leg Stand Test	4.54 (−4.31, 13.38)
MPA → Seated Up-and-Go Test	−0.60 (−1.22, 0.02)
HPA → Seated Up-and-Go Test	−1.12 (−1.69, −0.55) ***
Functional Fitness → Life Satisfaction	
Arm Curl Test → Life Satisfaction	−0.02 (−0.05, 0.01)
Chair Stand Test → Life Satisfaction	0.00 (−0.03, 0.04)
Back Scratch Test → Life Satisfaction	0.00 (−0.01, 0.01)
Chair Sit and Reach Test → Life Satisfaction	−0.00 (−0.02, 0.01)
2-Minute Step Test → Life Satisfaction	0.01 (−0.00, 0.01)
Single Leg Stand Test → Life Satisfaction	0.00 (−0.00, 0.00)
Seated Up-and-Go Test → Life Satisfaction	−0.04 (−0.10, −0.00)
Physical Activity → Functional Fitness → Life Satisfaction	
MPA → Arm Curl Test → Life Satisfaction	−0.00 (−0.03, 0.03)
HPA → Arm Curl Test → Life Satisfaction	−0.01 (−0.05, 0.01)
MPA → Chair Stand Test → Life Satisfaction	0.00 (−0.03, 0.04)
HPA → Chair Stand Test → Life Satisfaction	0.00 (−0.06, 0.08)
MPA → Back Scratch Test → Life Satisfaction	0.00 (−0.03, 0.04)
HPA → Back Scratch Test → Life Satisfaction	0.00 (−0.04, 0.05)
MPA → Chair Sit and Reach Test → Life Satisfaction	−0.00 (−0.03, 0.03)
HPA → Chair Sit and Reach Test → Life Satisfaction	−0.00 (−0.06, 0.05)
MPA → 2-Minute Step Test → Life Satisfaction	−0.01 (−0.06, 0.02)
HPA → 2-Minute Step Test → Life Satisfaction	0.03 (−0.01, 0.10)
MPA → Single Leg Stand Test → Life Satisfaction	0.01 (−0.03, 0.04)
HPA → Single Leg Stand Test → Life Satisfaction	0.00 (−0.05, 0.04)
MPA → Seated Up-and-Go Test → Life Satisfaction	0.07 (−0.01, 0.20)
HPA → Seated Up-and-Go Test → Life Satisfaction	0.10 (0.03, 0.29)

* *p* < 0.05, ** *p* < 0.01, *** *p* < 0.001.

## Data Availability

Some or all the data and models that support the findings of this study are available from the corresponding author upon reasonable request.

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
