# Peer review of "The Associations between Physical Activity, Functional Fitness, and Life Satisfaction among Community-Dwelling Older Adults"

_ijerph, 2022, doi:10.3390/ijerph19138043_

Round 1

Reviewer 1 Report

General comments

In this cross-sectional study, the authors aimed to explore the relationships between physical activity and functional fitness and life satisfaction, and the mediating role of functional fitness therein.

The topic and the findings are important. The authors found that promoting physical activity may be useful for increasing older adults’ functional fitness and life satisfaction, but the effect of physical activity on life satisfaction was not mediated by functional fitness.

I have some comments to make to the authors.

Specific comments

Abstract

1)     Line 19: After “623 older adults” add “(73.71 ± 5.91 years, M ± SD)”

Keywords

2)     Replace the keywords (Functional fitness; Life satisfaction; Physical activity) already present in the title with different ones to optimize the search for the manuscript through search engines.

Introduction

3)     Line 55-57: Remove the aim of the study and put it at the end of the Introduction

Methods

Participants, Study Design, and measurements are clearly explained. Instruments are validated.

4)     Why is the term PROCESS written in capital letters? (lines 133,138 and 214)

5)     Line 142: Replace with “p value <0.05”

Results

The tables and figures must be inserted immediately after their indication in the text and in the numerical order.

6)     After line 148 insert Table 1

7)     Lines 146-148: Remove the data already present in the table. Here you can write, for example: “A total of 623 older adults were recruited. The demographic characteristics of the participants are shown in Table 1.”

The information in the tables and figures can be supplemented in the text but the numbers already available in the tables and figures must not be repeated.

8)     After line 155 insert Table 2.

9)     Lines 149-155:  Remove the data already present in the table.

10)  After line 166 insert Table 3.

11)  Lines 156-166: Remove the data already present in the table.

12)  After line 167 insert Figure 2.

13)  Table 1: Replace “Age (Mean, SD)” with “Age(years, M ± SD)”

14)  Table 1: Replace “Mean=73.71(SD=5.91)” with “73.71 ± 5.91”

Discussion

This section is written clearly and to point. Limitations are described.

15)  Line 193: Remove “(ADL)”, it is not necessary because the acronym is not repeated later.

16)  Line 201: Replace “(2015)” with “[15]

17)  Line 201: Replace “(2016)” with “[5]

Conclusions

The conclusions are justified, and the take-home message is clear.

18)  Add what  direction  should  research  work  take  in  this  area  (future directions/implications). 

Reviewer 2 Report

The article "The associations between physical activity, functional fitness, and life satisfaction among community-dwelling older adults" is interesting, even if it doesn’t add much news on the subject; but in a world that continues to age more and more it is important to study the relationships and prospects for a healthy aging. However, there are still some things to arrange within the study.

First of all, the Introduction is too synthetic: needs to be expanded. in this way it is too hasty and unclear. Are we talking about the importance of the physical or mental health of the elderly? Or both? And why they could represent a potential problem? In addition, lines 55-56 are repeated in lines 67-68. 

I’m not convinced of the way satisfaction with life has been analyzed. The use of a questionnaire would not have given more information and more precise, than a single element?

Didn’t you think to analyze also the possible difference between man and woman?

In the end, even the abstract should be revised because it is not exhaustive, there is a lack of contextualization and you do not understand well what is studied in the article.

Reviewer 3 Report

Dear authors,

Congratulation on your paper. This study aimed to analyze the relationships between physical activity and functional fitness and life satisfaction, and the mediating role of function. Please consider the comments below some points to help the authors to improve the manuscript’ quality.

Overall, although the research topic is interesting and useful in the clinical field, I found the manuscript weak and there is a lack of information on how this study can be important for the clinical field. Authors should justify in the introduction the importance of performing this study rather than state only the purpose. As we can see there are a plethora of studies cited in the introduction. What is the new point of this study and how it can be useful for health professionals and the elderly?

The references do not follow the same style throughout the entire document. Please it must be corrected.

The discussion may benefit from stronger arguments and a better discussion of the findings. 

Introduction 

In Taiwan, older adults will comprise 20% of the whole population by 2025, a super-aged society.” Please, add a reference in this sentence. 

With the rise in this population, health promotion for this population is an important issue” You repeated the word population twice. Please change the sentence. 

“Physical activity is one of the many strategies to improve older adults’ health, and it has not only physical but also psychological benefits for older adults [2].” I would suggest re-writing to “Physical activity is one of the many strategies to improve older adults’ health, and it has physical and psychological benefits for older adults [2].”

However, some studies state the opposite, claiming that there were no relationships between physical activity and upper limb muscle strength [11, 12, 14], lower limb muscle strength [8, 11, 12, 14], upper limb flexibility [4, 48 8, 10, 14], lower limb flexibility [8, 10, 11, 14], cardiorespiratory endurance [8, 49 13, 14], and dynamic balance ability [11, 14].

Please, consider “Were” instead of “was” if you are talking about relationships (plural). I strongly recommend you to change to “(…) upper and lower limbs muscle strength and upper and lower limb flexibility… (…)“

Melo et al. are not in the same reference format as the others. The same for Blair, Cheng & Holder.

How this paper can be useful in the clinical field? In my opinion, there is a lack of justification for the practical applicability of this study. Why this manuscript will help professionals, health care settings, and the elderly? 

Methods

Why did the authors choose IPAQ for physical activity assessment in the elderly? 

Why did the authors seem to use random references for the functional fitness assessment? The tests look like the Senior Fitness Test battery, but the references were chosen randomly. 

In the 2-minute step test, did the authors count either the number of cycles or the number of times the knees raised to the middle distance between the iliac crest and patella? The original test includes the number of full steps completed in 2 minutes, raising each knee to a point midway between the patella and iliac crest (top hip bone). The score is the number of times the right knee reaches the required height. 

More description is needed in those tests. 

Why did the authors choose IPAQ for this study? Some explanation should be stated in the introduction. How the studies have been studying PA in older adults? 

Discussion

The findings of this study support that physical activity is correlated positively to most functional fitness categories, which is consistent with previous 191 studies [4-7, 9] and a review [28]” The reference 28 is a systematic review. So, the findings reported by the authors are consistent with previous studies.  There is no need for distinguishing between studies and the review.

Once again, the references in the discussion do not follow the same citation style. Example: Dondzila et al. (2015) and Duncan et al. (2016). 

Overall, the discussion has weak arguments and must be improved. 

Once again, there is a lack of a link between the finding and the practicability of this study. How/why this study can be important?

Round 2

Reviewer 2 Report

Thank you authors for improving the article, now it is ready for the publication.

Reviewer 3 Report

Thank you for addressing all my questions. I hope it was helpful for your manuscript quality.